# Breast cancer patient-derived scaffolds enhance the understanding of PD-L1 regulation and T cell cytotoxicity
Elena Garre [1,2] ✉, Sara Rhost[1], Anna Gustafsson[1], Louis Szeponik [3], Thais Fenz Araujo[1], Marianne Quiding-Järbrink[3], Khalil Helou[4], Anders Ståhlberg [1,5,6] & Göran Landberg [1,2] ✉

Recent advances as well as obstacles for immune-based cancer treatment strategies, highlight the notable impact of patient cancer microenvironments on the immune cells and immune targets. Here, we use patient-derived scaffolds (PDS) generated from 110 primary breast cancers to monitor the impact of the cancer microenvironment on immune regulators. Pronounced variation in *PD-L1* expression is observed in cancer cells adapted to different patient scaffolds. This variation is further linked to clinical observations and correlated with specific proteins detected in the cell-free PDSs using mass spectrometry. When adding T cells to the PDS-based cancer cultures, the killing efficiency of activated T cells vary between the cultures, whereas non-activated T cells modulate the cancer cell *PD-L1* expression to treatment-predictive values, matching killing capacities of activated T cells. Surviving cancer cells show enrichment in cancer stem cell and epithelial-to-mesenchymal transition (EMT) features, suggesting that T cells may not efficiently target cells with metastatic potential. We conclude that clinically relevant insights in how to optimally target and guide immune-based cancer therapies can be obtained by including patient-derived scaffolds and cues from the cancer microenvironment in cancer patient handling and drug development.

Cancer is a complex disease including numerous malignancy subtypes with pronounced heterogeneity between patients and within the cancer population. Specific cancer niches consist of malignant cells and varying microenvironments containing non-malignant cells such as fibroblasts, endothelial cells, and immune cells, as well as non-cellular components including extracellular matrix (ECM) and soluble cues. Immune cells recruited to the cancer microenvironment mediate both innate and adaptive immune responses, significantly affecting cancer progression and patient outcome[1]. Cytotoxic T cells are key players in anticancer immune responses, and the presence of tumor-infiltrating lymphocytes (TILs) has been linked to patient prognosis and treatment responses in several cancer types, including colorectal, ovarian and breast cancer[2–6]. Conversely, cancer cells can orchestrate immune evasion by expressing immune checkpoint ligands that reduce the function of cytotoxic T cells through checkpoint receptor interactions, as illustrated by the PD-1/PD-L1 pathway[7].

Different immunotherapy strategies have successfully improved the progression-free survival for cancer patients and are now essential components of the first-line treatment. However, only a fraction of patients benefit from this therapeutic approach. Recent studies have demonstrated the importance of the cancer microenvironment in influencing the outcome of immunotherapies, promoting different resistance mechanisms that impair the immune response. The ECM constitutes a physical barrier, and the presence of immunosuppressive cells can prevent T cells from homing into the cancer niche as well as driving T cell exhaustion and dysfunction[8]. Factors like hypoxia and acidity as well as competition for resources within the cancer microenvironment, further influence the immunosuppressive environment[9]. The cancer microenvironment may also harbor specific niches where cancer stem cells (CSCs) survive and thrive, leading to treatment resistance, including immunotherapies[10]. Collective data highlight the importance of understanding how patient-

[1]Department of Laboratory Medicine, Institute of Biomedicine, Sahlgrenska Academy, Sahlgrenska Center for Cancer Research, University of Gothenburg, Gothenburg, Sweden. [2]Department of Clinical Pathology, Sahlgrenska University Hospital, Gothenburg, Sweden. [3]Department of Microbiology and Immunology, Institute of Biomedicine, Sahlgrenska Academy, University of Gothenburg, Gothenburg, Sweden. [4]Department of Oncology, Institute of Clinical Sciences, Sahlgrenska Center for Cancer Research, Sahlgrenska Academy, University of Gothenburg, Gothenburg, Sweden. [5]Wallenberg Center for Molecular and Translational Medicine, University of Gothenburg, Gothenburg, Sweden. [6]Department of Clinical Genetics and Genomics, Sahlgrenska University Hospital, Gothenburg, Sweden. ✉e-mail: elena.garre.garcia@gu.se; goran.landberg@gu.se

specific cancer microenvironments influence cancer immune targeting approaches[11,12].

A major challenge for preclinical cancer models is their ability to accurately represent in vivo cancer growth and provide useful insights for cancer drug discovery. Key factors for proper modeling of cancer growth and spread in relation to immune properties include ECM characteristics as well as contributions from various stroma cells and immune cell infiltrations[13]. An additional level of complexity for cancer models is reproducing the heterogeneity on various levels present in individual patient cancer lesions[14]. 3D-based cellular models have been proven useful for studying mechanisms of cancer immune escape[15–17], but with the clear drawbacks of lack of a native microenvironment and the dependency of poorly defined animal-derived matrices[14,18,19].

New human-based models preserving cancer microenvironmental features of the cancer niche, such as patient-derived scaffolds (PDSs), have the capacity to better represent patient cancer-stroma interactions compared to available 3D-models. We have previously demonstrated that PDSs produced from decellularized primary cancer samples maintained the tumor-specific architecture and composition, revealing properties of the patient cancer microenvironment, including treatment responses[20–26]. Here, we have investigated how individual PDSs from a cohort of breast cancer patients influenced the immune response to adapted cancer cell lines. Different decellularized scaffolds repopulated with breast cancer cell lines triggered varying gene expression changes of molecules involved in T cell regulation in adapted cancer cells, further linked to the PDS-protein composition as well as clinical behaviors of the original cancer disease. In PDS co-cultures of cancer cells and T cells, we observed a PDS-dependent influence on the activation and killing capacities of the T cells as well as *PD-L1* expression in the cancer cells. Furthermore, activated T cells in PDS-cultures predominantly targeted and killed the differentiated sub-population of cancer cells, resulting in an enrichment of cancer stem cells with EMT and pluripotency features. These findings demonstrate that the immune regulating capacities of the individual tumor microenvironment is preserved in the PDS model and can be decoded by studying adapted cancer cells and the effect of T cell interactions. The data emphasize the translational capacity of the PDS model in evaluating the immune response and potentially predicting immunotherapy effects based on patient heterogeneity.

## Results

### Patient-derived scaffolds trigger differential gene expression of molecules involved in T cell regulation based on intrinsic properties of the original cancer

To investigate if different cancer microenvironments provided by the PDSs could trigger changes in immune modulating molecules in cancer cells, we analyzed the expression of the PD-1 ligands *PD-L1* and *PD-L2* genes in the breast cancer cell lines MCF-7 and MDA-MB-231, adapted and expanding in 110 and 84 breast cancer PDSs respectively (Experimental workflow in Fig. 1, Supplementary Table 1). *PD-L1* and *PD-L2* expression were detected in both cell lines propagated in PDS cultures. Notably, 15 MCF-7 PDS cultures induced *PD-L2* expression despite its absence in MCF-7 cells grown in 2D cultures (Fig. 2a, Supplementary Fig. 1, Supplementary Tables 2 and 3). Remarkably, the expression levels of *PD-L1* and *PD-L2* varied substantially between PDS cultures, with the MCF-7 cell line exhibiting more pronounced differences compared to the triple-negative MDA-MB-231 (Supplementary Table 3). This highlights a regulatory function for the human-based growth conditions, further influenced by the cancer cell line's response to microenvironmental cues.

Next, we analyzed how the PDS influence on the immune-related ligands were linked to available clinical information from the original breast cancer (Supplementary Table 1 and 4). Interestingly, *PD-L1* upregulation in MCF-7 PDS cultures was significantly associated with high-grade and the ductal subtype of breast cancer (Fig. 2b, c). There was also a tendency for an impaired disease-free survival for patients with higher induction of *PD-L1* in the PDS cultures ($p = 0.09$) (Fig. 2d). The *PD-L1* expression in MCF-7 cells were also compared with analyses delineating PDS-dependent expression changes in genes associated with EMT, proliferation, differentiation and cancer stem cell (CSC) features (Supplementary Table 5). Notably, *PD-L1* expression was significantly positively correlated with *SNAI2* ($p = 0.001$) and *ALDH1A3* ($p = 0.001$) expression, and all these PDS-induced changes were significantly linked to tumor grade[23]. Since the MDA-MB-231 cell line showed less responsiveness to the individual PDSs compared to MCF-7

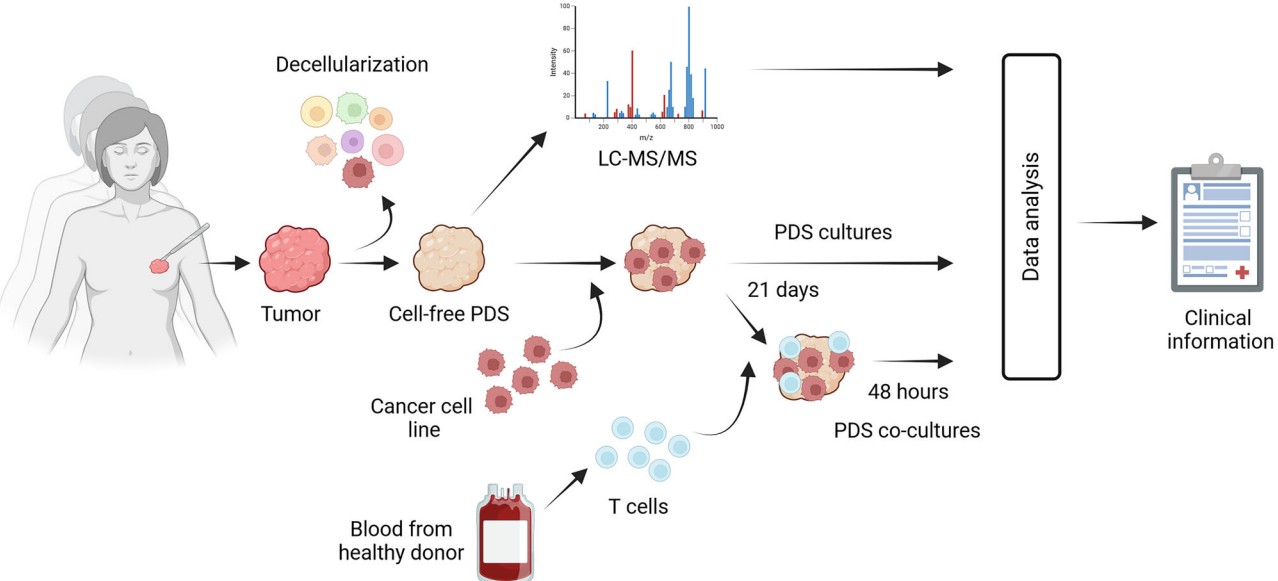

**Fig. 1 | Schematic experimental workflow illustrating the process from cancer surgery to data analyses of the patient-derived scaffold (PDS).** Primary cancer samples are decellularized to generate cell-free PDSs that can be homogenized, and protein content analyzed using quantitative LC-MS/MS; or can be used for cancer cell line cultures. After 21 days of cancer cell adaptation to the patient cancer microenvironment, T cells isolated from healthy donor blood can be added and incubated for 24-48 hours (PDS co-cultures). Different analysis of the PDS cultures can be performed to characterize the cell responses to the tumor microenvironment provided by the PDSs and compared with clinical information. Created in BioRender (https://BioRender.com/v80h354).

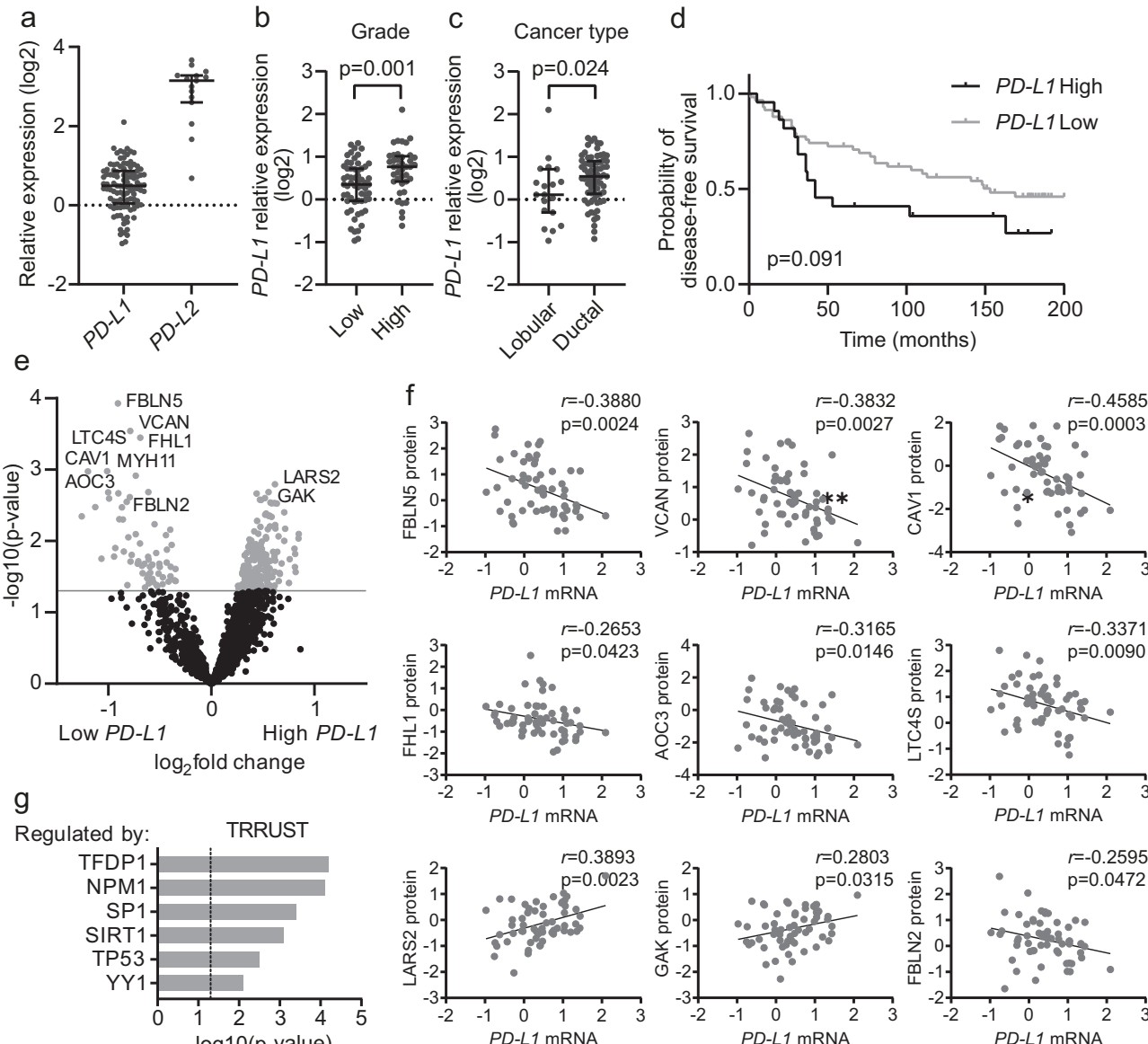

**Fig. 2 | PDS composition triggers *PD-L1* gene expression, which is associated with clinical features of the original cancer. a** Dot plot showing changes in *PD-L1* and *PD-L2* gene expression in patient-derived scaffolds (PDS). Individual dots represent relative gene expression of an individual PDS grown with MCF-7 (n = 110 independent PDSs) to gene expression of MCF-7 cultured in 2D (log2-scale). Median and interquartile range (IQR) represented by errors bars are plotted (Supplementary Table 3). Dot plots including median and IQR, representing PDS-induced *PD-L1* gene expression in MCF-7 showing significant association to the clinical variables grade (**b**) and cancer type (**c**)(*p* < 0.05). **d** Kaplan–Meier plot displaying the relationship between *PD-L1* low (grey line) and high (black line) expression PDS-induced and disease-free survival (Log-rank statistical test and third

quartile, Q3, for data stratification were applied). **e** Volcano plot depicting the 247 proteins differentially expressed in 59 cell-free PDSs inducing low and high *PD-L1* expression in MCF-7 cancer cells. Significant proteins are considered as *p* < 0.05 in unpaired two-sided Welch t-test and are colored grey. The 10 most varying proteins are indicated. **f** Scatter plots illustrating significant correlations between protein levels of the most differentially expressed proteins in cell-free PDSs and *PD-L1* mRNA levels induced in MCF-7 cancer cells grown on them (log2-scale). Pearson's correlation coefficients (*r*) and the significance (*p* < 0.05) are indicated. **g** Bar graph depicting TRRUST analysis showing the most enriched human transcription factor interactions for the 247 differentially expressed proteins in the cell-free PDSs.

cells, and the gene changes were not associated with clinical observations (Supplementary Table 4), we focused on the more informative MCF-7 cancer cell line for the upcoming PDS experiments.

Subsequently, we investigated how proteins from the cancer microenvironment retained in the PDSs were associated with *PD-L1* expression changes in adapted cancer cells, stratified according to low or high *PD-L1* inducing capacity in MCF-7 cancer cells (Fig. 2d, Supplementary Table 1). 247 proteins were significantly differently expressed between the groups, including 65 proteins enriched in *PD-L1* low-inducing PDSs and 182 proteins more abundant in *PD-L1* high-inducing PDSs (Fig. 2e, Supplementary Data 1, Supplementary Fig. 2). Moreover, 9 out of the 10 most differentially

expressed proteins correlated significantly with the *PD-L1* mRNA changes in MCF-7 cancer cells (Fig. 2f, Supplementary Fig. 3). Interestingly, 7 out of the 10 proteins had a direct or indirect association with T cell and inflammatory functions (CAV1, VCAN, LTC4S, MYH11, AOC3, LARS, GAK) (Supplementary Data 1). Further, genes coding for the 247 identified proteins were enriched for transcription factor targets involved in *PD-L1* regulation, such as NPM1, SP1, SIRT1, TP53 and YY1 (Fig. 2g). Altogether, the presented data demonstrate that the PDS-dependent *PD-L1* expression changes in MCF-7 cancer cells were related to the specific protein composition of individual PDSs, including proteins with immune regulatory functions.

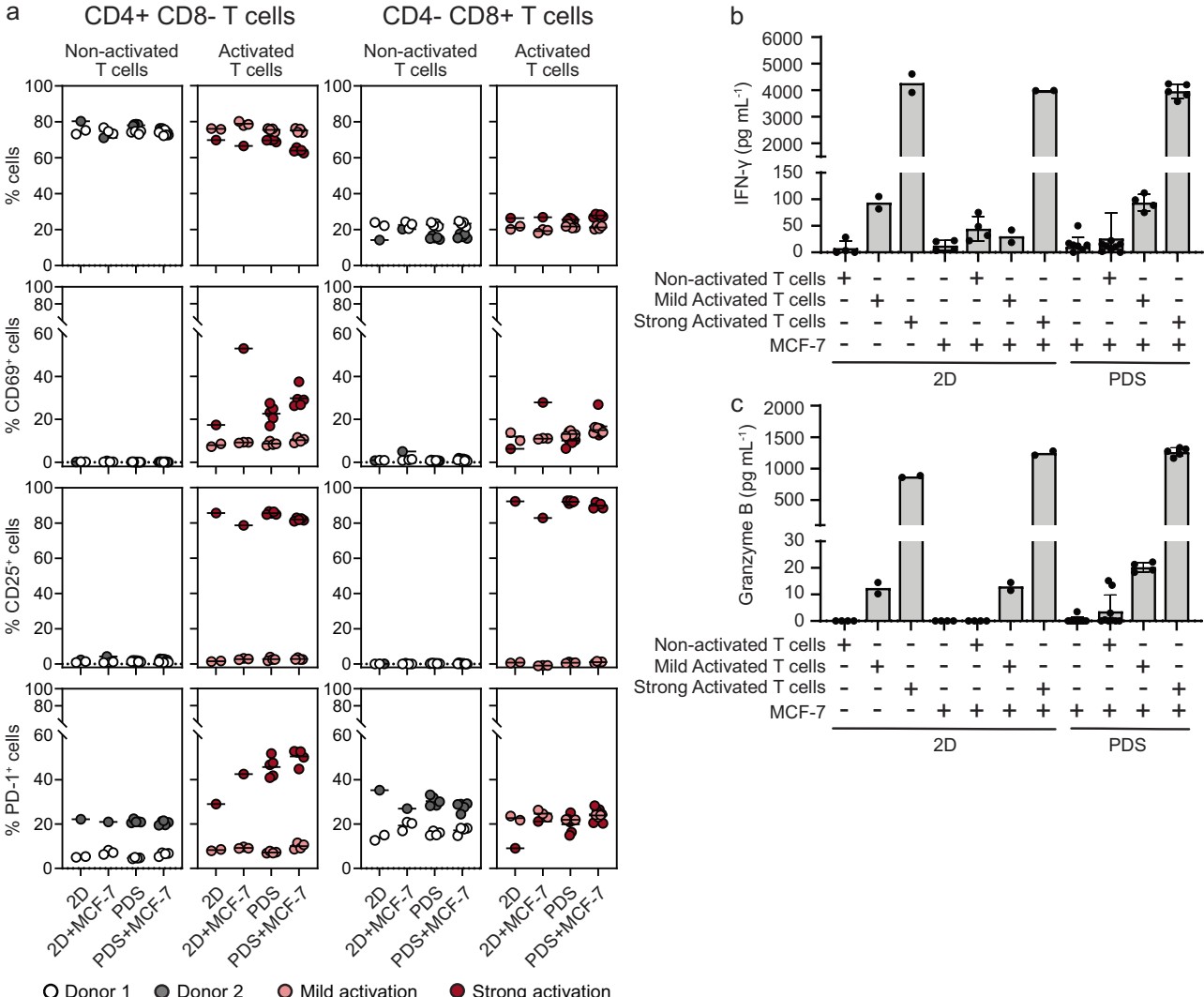

**Fig. 3 | PDS contribution to the activation of the T cells. a** Percentage of CD3+CD4+CD8- or CD3+CD4-CD8+ cell subsets expressing CD69, CD25 and PD-1 markers in non-activated T cells (white dots, donor 1; grey dots, donor 2) and after mild (light red dots) or strong activation (dark red dots). T cells were cultivated in 2D or PDSs and in mono- or co-cultures with MCF-7/Luciferase cancer cells. Two different T cell batches isolated from two different healthy donors were used. Mean and standard deviation (SD) are plotted (*n* = 1-5 independent replicates). IFN-γ (**b**) and Granzyme B (**c**) released to media in 2D or PDS mono- and co-cultures of MCF-7 cancer cells and T cells (pg mL⁻¹). Mean and standard deviation (SD) are plotted (*n* = 2-9 independent replicates).

## PDSs contribute to the activation of T cells increasing the fraction of CD69 and PD-1 positive cells

Considering the profound effects of PDS cultures on cancer cells, we hypothesized that the PDS could influence T cell properties as well the interaction between T cells and adapted cancer cells. T cells isolated from healthy donors, were therefore activated (mild or strong activation) and added to PDS cultures of MCF-7/Luciferase cancer cells established for 21 days, and co-cultured for 24 or 48 hours (Experimental workflow in Fig. 1). Control 2D or PDS cultures including cancer cells or T cells in monocultures and co-cultures with non-activated T cells, were propagated in parallel. Flow cytometric analyses of T cells from the different culture conditions showed similar frequencies of CD4+ and CD8+ T cells independently of the culture format or presence of MCF-7/Luciferase cancer cells (Fig. 3a). The activation level of the T cells monitored by analyses of the surface markers CD69, CD25 and PD-1, further showed increasing fractions of CD69+ and CD25+ T cells corresponding to the activation protocol. Interestingly, cell-free PDSs boosted the frequency of CD69+ T cells in both strongly activated CD4+ and CD8+ populations, and the frequencies were even higher in co-cultures with cancer cells. The T cell activation protocols stimulated the surface expression of PD-1, with a significant increase in the

CD4+ population after strong activation. Moreover, for strongly activated T cells, the frequency of PD-1+ cells increased in the PDS cultures with or without MCF-7/Luciferase cancer cells. The secreted anti-tumor immune effector molecules interferon-gamma (IFN-γ) and granzyme B increased in the media from cultures including active T cells, with a pronounced boost using the strong activation protocol, and even higher for Granzyme B in PDS co-cultures, agreeing with the phenotypes observed by flow cytometry (Fig. 3b, c).

## PDS-based co-cultures influence the *PD-L1* expression by cancer cells as well as the killing capacity of the T cells

The killing capacity of T cells in PDS cultures was analyzed by defining the amount of surviving MCF-7/Luciferase cancer cells in co-cultures with or without T cells using a luminescence assay (Fig. 4a, b). The addition of non-activated T cells or mildly activated T cells to MCF-7/Luciferase cancer cells growing in 2D, did not cause a significant decrease in the number of cancer cells, whereas cancer cell numbers dropped significantly when adding strongly activated T cells. In the PDS co-cultures including non-activated T cells, there were no significant changes in cancer cell survival compared to PDS cultures without T cells, even though there was

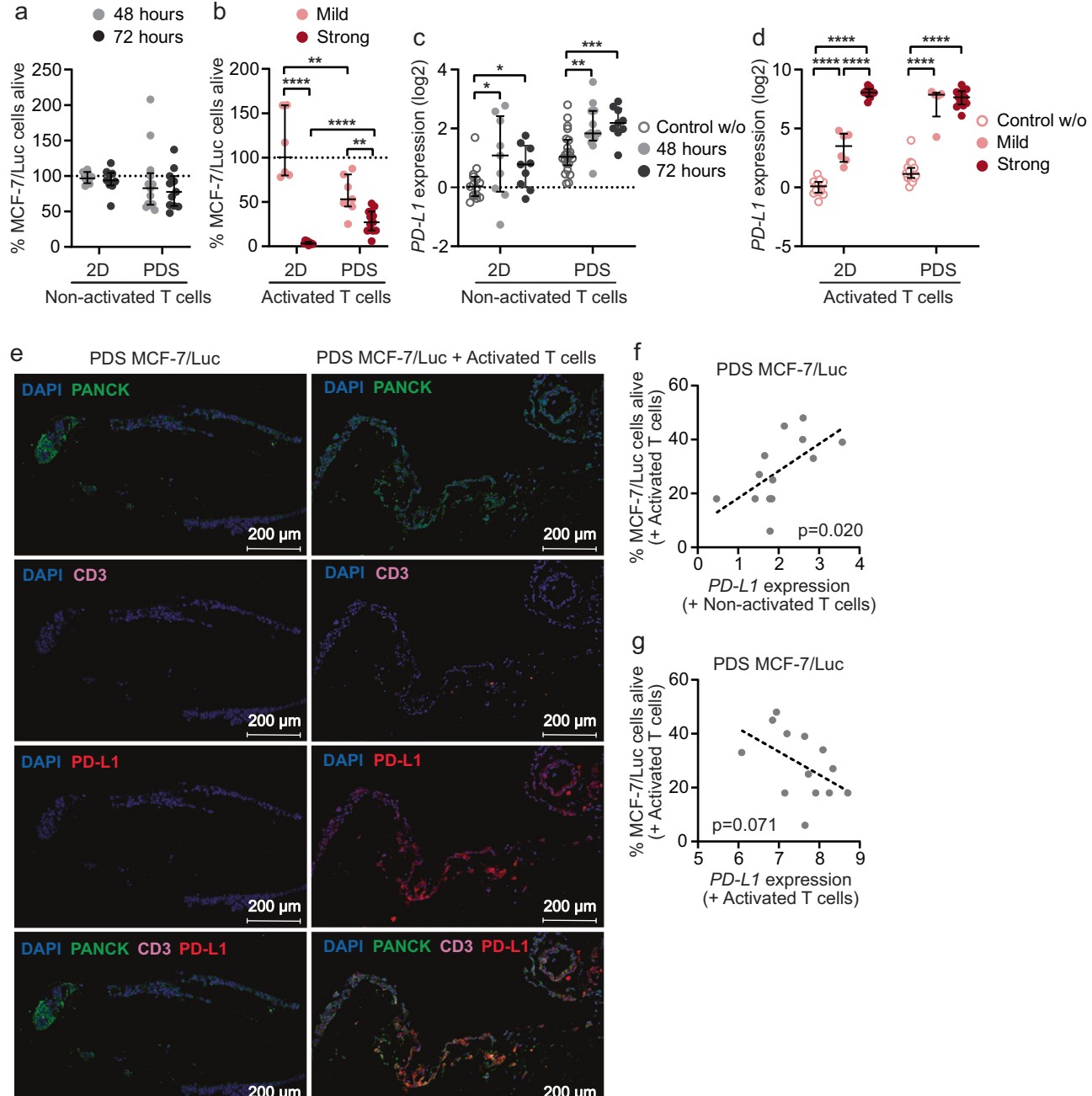

**Fig. 4 | MCF-7 cancer cell viability and *PD-L1* expression in PDS co-cultures with T cells.** Percentage of alive MCF-7/Luciferase cancer cells (**a**, **b**) and *PD-L1* gene expression (**c**, **d**) after incubation with non-activated (**a**, **c**) (grey dots, 48 hours; black dots, 72 hours) or activated T cells (**b**, **d**) in 2D and PDS cultures. T cells were activated to two levels, mild (light red dots) and strong (dark red dots). 2D and PDS cultures with only MCF-7/Luciferase cells were used as 100% viability reference and included for comparison in gene expression analyses. Each PDS culture is represented by an individual dot (*n* = 5–22 independent PDS cultures), and median and interquartile range (IQR) represented by errors bars are plotted (Supplementary Table 6 and 7). Significant differences are analyzed using unpaired t-test, two-tailed ($^*p < 0.05$, $^{**}p < 0.01$, $^{***}p < 0.005$, $^{****}p < 0.0001$). **e** PDS sections with only MCF-7 cancer cells or in co-culture with strongly activated T cells showing cancer cells (PanCK[+], green), T cells (CD3[+], pink) and PD-L1 expression (red). Nuclei were stained with DAPI (blue). Scatter plots illustrating correlations between the percentage of alive MCF-7/Luciferase cancer cells in PDSs (*n* = 13 independent PDSs) after incubation with strongly activated T cells and *PD-L1* expression in presence of non-activated T cells (**f**) or strongly activated T cells (**g**). Individual dots represent each PDS co-culture. Spearman's correlation significance ($p < 0.05$) is indicated (Supplementary Table 8).

a marked variability between the PDS cultures reaching a 50% reduction of cancer cells in some of them (Fig. 4a; Supplementary Table 6). There was also a gradual decrease in the number of MCF-7/Luciferase cells in PDS co-cultures related to the activation grade (53% and 27% median values of surviving cells with mild and strong activation respectively; Fig. 4b). Interestingly, the mild activation of T cells resulted in a clear

increased killing-effect in PDS co-cultures compared to 2D cultures, whereas the effect of the strong activation was less exacerbated in PDS cultures compared to 2D cultures (Fig. 4b). Different PDS co-cultures with cancer cells and T cells, further showed varying cancer cell survival, underlining that the heterogeneity of the scaffolds and human-based cancer microenvironments influenced the effect of activated T cells.

As previously observed (Fig. 2), PDS growth and adaptation to a patient-based cancer microenvironment markedly induced *PD-L1* expression in the MCF-7/Luciferase cells (Fig. 4c, d). The presence of non-activated T cells also contributed to *PD-L1* mRNA levels in PDS-adapted cancer cells (Fig. 4c), while the most pronounced increase was observed for co-cultures using activated T cells even at mild activation (Fig. 4d, Supplementary Table 7). Immunofluorescence staining corroborated that the PD-L1 protein levels in MCF-7/Luciferase cells was lower when growing solitary but increased in the presence of activated T cells (Fig. 4e). Interestingly, the induction of *PD-L1* expression in the cancer cells by the individual PDSs in presence of non-activated T cells was significantly linked to the killing effect of activated T cells in parallel PDS co-cultures (Fig. 4f, Supplementary Table 8). These data illustrate that a PDS complemented with non-activated T cells representing a more comprehensive cancer niche, induced *PD-L1* levels in cancer cells matching treatment predictive values for T cell killing capacities. In addition, the surviving cancer cells in the PDS-cultures incubated with strongly activated T cells, showed a tendency towards high expression of *PD-L1* (*p* = 0.071) (Fig. 4g). Altogether, the results indicated that the exposure of MCF-7/Luciferase cells to activated T cells promoted the enrichment of PD-L1 high expressing cancer cells, probably due to a combination of increased *PD-L1* gene expression and selection of PD-L1 positive cancer cells not susceptible to be killed by the T cells.

### PDS co-cultures with active T cells enrich for poorly differentiated MCF-7 cancer cells with high pluripotency capacity

Next, we used the PDS-based co-culture system, to define if specific subpopulations of cancer cells were differentially targeted by the T cells in human-based microenvironments (Supplementary Table 9). The surviving subpopulations of cancer cells were characterized by analyzing the expression of genes associated to relevant cancer processes such as differentiation, proliferation, pluripotency, EMT and CSC features (Fig. 5a, Supplementary Fig. 5). PDS co-cultures including non-activated T cells did not induce any major changes in the cancer cells for the monitored processes, whereas the addition of activated T cells clearly modified the gene expression pattern of the surviving cancer cells compared to cultures without T cells. PDS cultures with mildly activated T cells caused a pronounced upregulation of EMT genes *VIM2* and *SNAI1* in the MCF-7 cancer cells, while co-cultures with strongly activated T cells resulted in increased expression of pluripotency genes *POU5F1*, *NEAT1* and *NANOG* as well as CSC-related gene *CD44* and downregulation of differentiation genes *ESR1* and *CD24*, besides an increased expression of the EMT markers including *FOSL1*. Similar results were observed when CD3$^+$ cells were depleted from the samples before RNA extraction, indicating that the main observations were derived from cancer cells (Supplementary Fig. 6). Altogether, the results indicated that activated T cells mainly targeted differentiated and proliferative cancer cells in the human-based cultures conditions using PDS, leaving less differentiated cancer cells enriched for mesenchymal, pluripotency and stemness features.

### Cancer cell susceptibility to T cell killing and PD-1 targeted immunotherapy is PDS-dependent and associated with clinical properties

A strength of the PDS model is the preserved patient-specific protein composition of the cancer microenvironment, capable of inducing adaptions of the cultured cancer cells towards original cancer features mimicking the patient-specific cancer niche (Fig. 2). To further delineate the relevance of modeling cancer "outside the patient", we compared the data from 12 PDS co-cultures including MCF-7/Luciferase cancer cells and strongly activated T cells with the clinical information from the original breast cancers (Supplementary Table 9). Interestingly, there was a significant link between the proliferation of the primary cancer cells (% Ki-67) and the killing capacity of the activated T cells towards adapted MCF-7 cancer cells, suggesting that low proliferative breast cancers might be less sensitive to T cell killing (Fig. 5b, Supplementary Table 10). We next analyzed if the phenotypes of the adapted MCF-7 cancer cells to the individual PDSs were

linked to the susceptibility to T cell killing. Several gene expression changes in PDS-adapted cancer cells prior to T cell addition, were significantly linked to the fraction of surviving cancer cells after incubation with T cells (Schema in Supplementary Fig. 7, Supplementary Table 11). Specifically, expression changes in *SNAI1*, *CD44* and *ALDH1A3* genes were positively associated with the viable cancer cell fraction (rs > 0.5 and *p* < 0.05), suggesting that PDS cultures promoting CSC features might be less susceptible to T cells action (Fig. 5c, 0). In line with these data, PDS co-cultures with high viabilities after T cell addition ( + T cells), showed more pronounced expression of EMT genes (*CDH2*, *TWIST*, and *VIM2*) and the CSC gene *ABCG2* in the surviving cancer cell population (Fig. 5c; rs > 0.5 and *p* < 0.05).

As reported above, PDS co-cultures with active T cells were associated with an enrichment of *PD-L1* expressing cancer cells (Fig. 4). We therefore characterized how the T cell mediated cancer cell killing was affected by a PD-L1/PD-1 inhibitory signaling using the PD-1 blocking antibody Pembrolizumab in 10 different PDS co-cultures. Strongly activated T cells were incubated with 100 μg mL$^{-1}$ of Pembrolizumab for 30 min prior addition to the co-cultures, and the blocking efficiency was verified by flow cytometry (Supplementary Fig. 8a). Interestingly, the addition of activated T cells induced the expression of *PD-L1* in MCF-7/Luciferase independently of the PD-1 blocking (Supplementary Fig. 8b). Pembrolizumab treatment of the T cells in PDS co-cultures further showed varying effects on the T cell killing. Two PDS cultures, PDS4 and PDS10, were clearly defined as "responders," showing a 50-60% additive reduction in cancer cell viability. Three additional "responders," PDS2, PDS3, and PDS8, exhibited slightly lower treatment effects with viability reductions of 28%, 26%, and 11%, respectively. The remaining five PDS cultures showed only minor effects, with less than a 10% reduction in viability, and were classified as "non-responders" (Fig. 5d). Although there were few differences in gene expression in surviving cancer cells after Pembrolizumab treatment, PDS cultures not-responding to the treatment had significantly higher *SOX2* and lower *MKI67* expression compared to responding PDS cultures (Fig. 5e, f, Supplementary Fig. 8b, Supplementary Table 12). These data suggest an enrichment of cancer cells with pluripotency features and low proliferative phenotype in PDSs less susceptible to the PD-1 blocking.

## Discussion

One of the most challenging aspects of cancer modeling is the inclusion of the tumor-stroma interaction complexity and the heterogeneity between patients on multiple levels. By using patient-derived scaffolds (PDS) as a base for complex co-cultures complemented with various relevant cell types within the cancer niche, we can reveal critical details about functional interactions and effects in genuinely human-based models. Immune targeting therapies are highly relevant in modern cancer treatment schedules and breast cancer PDS cultures complemented with T cells, permit detailed studies of how critical immune regulating treatment targets as PD-L1[27], are affected by growth in comprehensive cancer models outside the patient.

Based on earlier publications reporting phenotypic changes of cancer cells adapted to PDSs[20,23], it is also likely that immune regulating processes will be affected by human-based cancer microenvironments[28]. Interestingly, individual PDS cultures indeed induced patient-specific changes in the *PD-L1* expression in the adapting breast cancer cell lines, further associated with fundamental clinico-pathological characteristics of the original cancer from where the PDS was obtained. Increased expression of *PD-L1* in scaffold cultures was significantly correlated with high grade and ductal breast cancers, as has previously been described for primary breast cancer lesions[29–32]. There was also a trend that high *PD-L1* induction by a PDS was associated with shorter disease-free survival for the donor cancer patient, unlike observations in primary cancer samples[31,33,34]. The associations between PD-L1 expression and prognosis in breast cancer nevertheless remain controversial and may potentially be influenced by several cell types expressing PD-L1 within the cancer niche, as well as misleading analyses of functional protein levels[32,35,36]. Our data observing a striking modulation of *PD-L1* mRNA levels depending on the tumor microenvironment, supports that relevant PD-L1 measurement for certain cancer patients needs to be

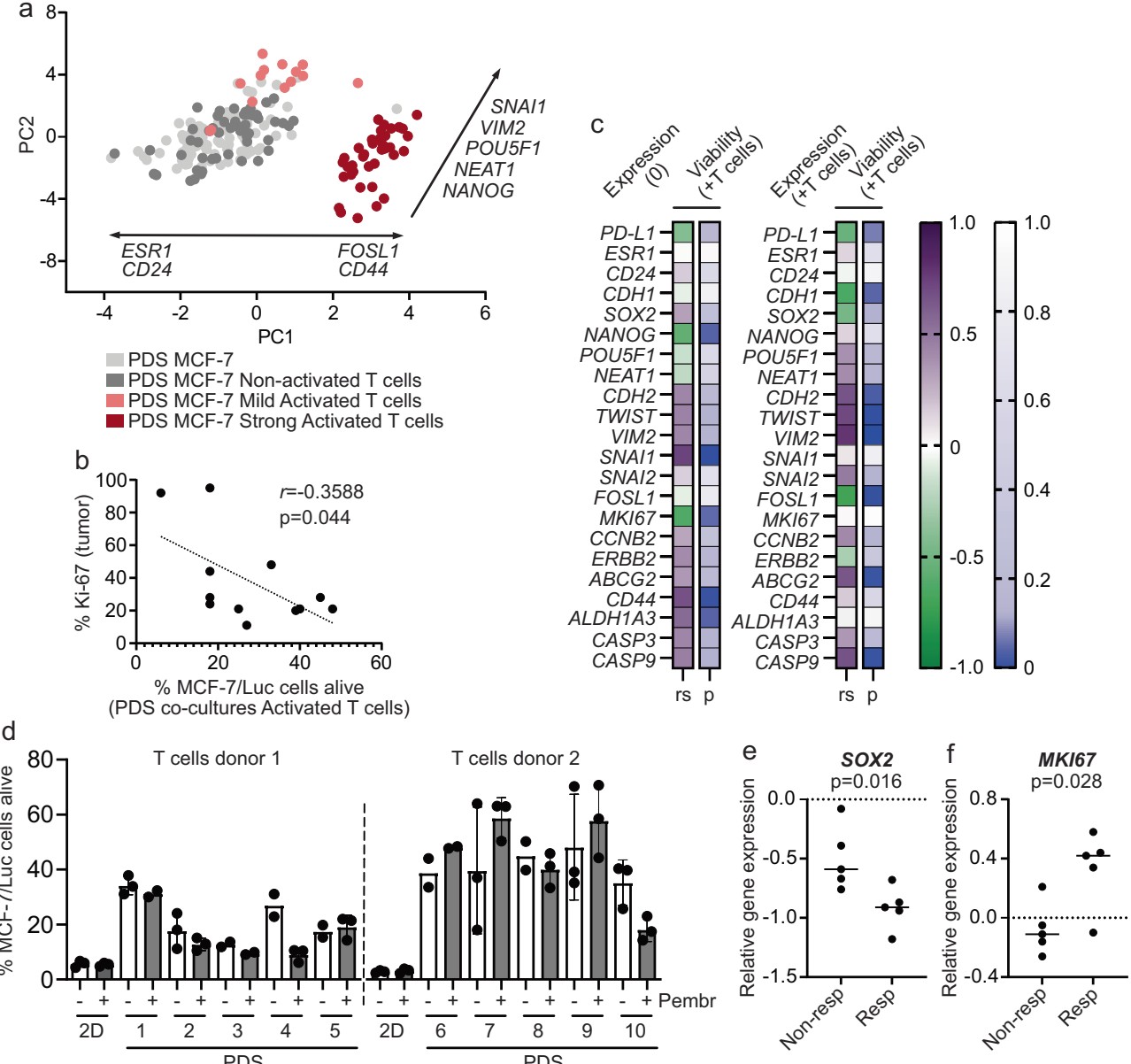

**Fig. 5 | Gene expression and cancer cell susceptibility to T cell killing are influenced in PDS-dependent manner. a** Principal component analysis (PCA) showing the distribution of PDSs cultured with MCF-7 alone (light grey dots) and with non-activated (dark grey dots), mildly activated (light red dots) or strongly activated (dark red dots) T cells (n = 17 independent PDSs, replicates = 3–6), based on the expression of 17 genes associated with relevant cancer processes. Genes contributing to the PDS distribution directionality are indicated. **b** Scatter plot illustrating the correlation between the percentage of alive MCF-7/Luciferase cancer cells in PDSs after incubation with strongly activated T cells and the Ki-67 expression histologically analyzed in the original tumor. Individual dots represent each PDS (n = 12 PDSs). Pearson's correlation coefficient (r) and significance (p < 0.05) are indicated (Supplementary Table 10). **c** Heat map depicting Spearman correlation analysis (rs, correlation value; p, p-value) between MCF-7/Luciferase cancer cell gene expression determined

before (0) and after ( + T cells) addition of strongly activated T cells versus cancer cell viability in PDS co-cultures with strongly activated T cells. **d** Percentage of alive MCF-7/Luciferase cancer cells in PDS co-cultures (n = 10 independent PDSs) after incubation with strongly activated T cells (isolated from two different healthy donors) in absence (white bars) or presence (grey bars) of 100 µg mL⁻¹ Pembrolizumab (Pembr). 2D co-cultures were included as control and cultures with only MCF-7/Luciferase cells were used as 100% viability reference. Mean and standard deviation are shown (n = 2-3 independent replicates). Dot plots depicting the SOX2 (**e**) and MKI67 (**f**) gene expression in MCF-7 cancer cells in PDS cultures that responded (Resp) or not (Non-resp) to Pembrolizumab treatment. Each PDS culture is represented by individual dots (n = 10 independent PDSs), median is plotted, and significance indicated (p < 0.05) (Supplementary Table 12).

determined in more complex growth models better mimicking in vivo conditions.

As previously reported, the composition of the cell-free PDSs is complex and includes not only ECM-associated proteins but also proteins related to secretion, immune response, and metabolism, reflecting the previous presence and activities of different cells in the primary cancer niche[20,37]. When analyzing the protein composition of cell-free PDSs,

grouped by PD-L1 expression induction in adapting cancer cells, we observed an enrichment of proteins regulated by transcription factors involved in PD-L1 gene expression regulation as well as immune escape[38–44]. Moreover, PDSs linked to low PD-L1 expression were enriched for proteins associated with immunosuppressive TME and T cell exclusion such as VCAN1, FHL1, MYH11, LTC4S and AOC3, and efficient CD8⁺ T cell expansion and effector function such as Caveolin-1 (CAV1), suggesting two

scenarios within the primary cancer with low T cell infiltration or an effective cytotoxic T cell infiltration and response[36,45–53]. In contrast, LARS and GAK proteins related to T cell apoptosis and inhibition of IL-12 mediated IFNγ production were linked to high *PD-L1* inducing activities in the PDS cultures, in line with a primary cancer niche including T cell infiltration but suppressed cytotoxic activity[54,55]. The data support that the patient-specific composition and structure of the cell-free PDS, indeed promote phenotypic changes of the introduced cancer cells indicative of the complex immune microenvironment in the primary cancer niche.

With this study, we have illustrated that the cancer microenvironment preserved in the PDS, promotes a distinctive cancer cell – T cell interaction not present in monolayer cultures by influencing T cell activation, increasing the percentage of CD69+ and PD-1+ T cells, contributing to *PD-L1* induction in MCF-7 cancer cells and affecting the cancer cells susceptibility to T cell killing, including the efficacy of the immune checkpoint inhibitor Pembrolizumab. Importantly and as discussed above, many proteins related to immune response modulation were retained in the PDSs affecting the adaptation of cancer cells as well as T cells, and also influencing alternative immune checkpoint signaling components[56]. This differential protection against T cell killing, might be partially explained by a distinct stimulation of antiapoptotic signal in the cancer cells or dysfunction of activated T cells mediated by PD-L1/PD-1 axis[36,57–59]. Moreover, the actual ECM-components of the PDSs could also impact the migration and activity of T cells depending on differences in collagen density and fiber organization and scaffold composition[60,61].

Recent studies have indicated a strong inverse association between stemness and immunity signatures across several cancer types, including resistance to T cell attack[12,62,63]. PD-L1 has also been reported to be highly expressed in cancer stem cells (CSC) in numerous cancer types and further associated with the promotion of stemness functions[64–67]. In breast cancer, PD-L1 can promote cancer cell stemness by supporting the expression of the stem cell master transcription factors Oct-4 (*POU5F1* gene) and Nanog[64]. Consistent with these observations, we detected an enrichment of cancer cells with high *PD-L1* expression in co-cultures with activated T cells, parallel to increased expression of *POU5F1* and *NANOG* genes. Surviving MCF-7 cancer cells further showed high *CD44* and low *CD24* gene expression, a cancer stem cell phenotype that has been described to be promoted by polyclonal allogenic T cells[68,69]. The presented data align with previous studies relating CSC with PD-L1 expression and resistance to T cell attack, highlighting the relevance of targeting cancer stemness features in combination with immunotherapies to increase the overall treatment efficiency.

Besides a general representation of human cancer microenvironments, patient-derived scaffolds also provide exclusive insights to the primary cancer niche for a specific patient. Here, we used this unique attribute to analyze how individual cancer microenvironments influence cancer - T cell interactions. Our data also supported that PDS cultures mirror patient specific and clinically relevant features. Transcriptomic and computational studies have highlighted the relevance of the immune contexture for patient outcomes, showing proliferative breast cancer phenotypes accompanied with pronounced infiltration of antitumoral cell types, and further related to better treatment response[70,71]. Meanwhile, EMT and stem cell phenotypes have predominantly been associated with immunosuppressive features and adverse patient outcomes[70]. In line with the reported studies, our data indicated a negative correlation between the amount of surviving cancer cells in PDS cultures with activated T cells, and the proliferative capacity of the original breast cancer; and that PDSs promoting CSC and EMT features, were less susceptible to T cell killing. Furthermore, PDS-dependent Pembrolizumab responses with lower T cells killing capacities, were strongly associated with high expression of *SOX2* gene in cancer cells, in agreement with a role for SOX2 in the maintenance of CSC features and the tumor-intrinsic mediated resistance to immune checkpoint blockade[72–74]. Given that a high percentage of the PDSs in this study originated from estrogen receptor-positive breast cancers, our data support previous studies indicating the relevance of the tumor microenvironment and immunological

state in hormone-positive breast cancer. This subtype may potentially benefit from immunotherapies within an individualized, predictive treatment strategy[75,76]. Although, estrogen receptor-positive breast cancer has not traditionally been considered immunogenic, varying levels of tumor-infiltrating lymphocytes (TILs) have been observed and linked to prognostic values[77]. Patients with aggressive metastatic hormone-positive breast cancer, have also been reported to respond to immune-checkpoint inhibitor treatment, although the effectiveness remains unclear[75,78]. Consistent with this, our results using PDS cultures indicated that some estrogen receptor-positive breast cancer microenvironments, including an estrogen receptor breast cancer cell line, were sensitive to Pembrolizumab. This provides an additional tool for evaluating and predicting patient-specific immunotherapy responses.

By using patient-derived scaffolds and adapting breast cancer cells sensing the various cues from the cancer niches, we have illustrated pronounced and varying effects on the clinically relevant cancer treatment target PD-L1. Besides, mass spectrometry identified several proteins in cell-free scaffolds associated with the *PD-L1* gene expression changes, having the potential to be targeted in novel combination treatment strategies increasing the immune therapy efficiency. Interestingly, the PDS-environments also influenced the killing capabilities of activated T cells, illustrating the importance of including human cancer microenvironments in the validation of T cell effects on cancer cells. In conclusion, this comprehensive human breast cancer-based study has highlighted clinically relevant insights in how key targets for immune therapies in cancer are affected by the cancer microenvironment.

## Material and methods
### Cell lines and culture conditions
MCF-7 and MDA-MB-231 breast cancer cell lines (ATCC; Manassas, VA, HTB-22™ and HBT-26™ respectively) were directly obtained from ATCC and authenticated by the manufactures' company. Cell lines were cultured in accordance with ATCC recommendations. MCF-7 cells were cultured in Dulbecco's modified Eagle's medium (DMEM) supplemented with 10% fetal bovine serum (FBS), 1% penicillin/streptomycin, 1% L-glutamine, 1% Antibiotic-Antimycotic (all Thermo Fisher Scientific) and 1% MEM Non-Essential Amino Acids (Sigma-Aldrich); while MDA-MB-231 cells were cultured in RPMI-1640 medium supplemented with 10% FBS, 1% penicillin/streptomycin, 1% Antibiotic-Antimycotic, 1% sodium pyruvate and 1% L-glutamine (all Thermo Fisher Scientific). MCF-7/Luciferase (Puromycin) breast cancer cell line (GenTarget Inc, United States) was cultured in RPMI-1640 medium (1% penicillin/streptomycin, 1% sodium pyruvate, 1% Antibiotic-Antimycotic and 1% L-glutamine) and 10% of heat-inactivated FBS. All cultures were performed at 37 °C with 5% $CO_2$ humidified atmosphere, and cell media was renewed every 3-4 days. Cell lines were confirmed as mycoplasma-free (Mycoplasma PCR Detection Kit, Applied Biological Materials Inc., Richmond, BC, Canada).

### Isolation of T cells and activation methods
Peripheral blood mononuclear cells (PBMCs) were isolated using Ficoll–Paque (GE Healthcare) density gradient separation from healthy donor buffy coats. Buffy coats were purchased from a local blood bank (Sahlgrenska University Hospital). T cells were negatively selected using Dynabeads® Untouched™ Human T Cells kit (Invitrogen, Life Technologies). Two types of T cell activation were used: for mild activation, T cells were treated with immobilized anti-CD3 (0.1 µg mL⁻¹; clone OKT3; ref. 317325; Biolegend) and soluble anti-CD28 (0.1 µg mL⁻¹; clone CD28.2; ref. 302933; Biolegend) in RPMI-1640 medium (1% penicillin/streptomycin, 1% sodium pyruvate and 1% L-glutamine) supplemented with 10% heat-inactivated FBS for 20 hours and afterwards the antibodies were washed away; and for strong activation, T cells were incubated with Dynabeads® Human T-Activator CD3/CD28 (Gibco, Life Technologies) for 20 hours following the manufacturer's instructions. When the experiment required, T cells (1 × 10⁶ cells mL⁻¹) were incubated immediately after activation and before downstream steps, with 100 µg mL⁻¹ Pembrolizumab (Selleckchem) for 30 min.

## Patient material

Breast cancer tissues from female patients between 22-88 years old and without neoadjuvant therapy were collected from Breast Cancer Biobank or directly after surgery at the clinical pathology diagnostic unit at Sahlgrenska University Hospital (Gothenburg, Sweden). 110 frozen tumors from the Biobank (samples obtained from 1992 to 1999 and followed up until 2012) and 17 tumors directly after surgery were used in this study (Supplementary Table 1; Supplementary Table 9). Clinico-pathological characteristics and overall survival data were obtained from the National Quality Registry at the Regional Cancer Center West (Gothenburg, Sweden) and the Cancer Registry at the National Board of Health and Welfare, respectively. All ethical regulations relevant to human research participants were followed in accordance with the Declaration of Helsinki, and approved by the Regional Research Ethics Committee (Regionala Etikprövningsnämnden) in Gothenburg (DNR: 515-12, S164-02 and T972-18). Informed consent was obtained from all the participants involved in the study.

## Tumor decellularization and patient-derived scaffolds generation

Breast cancer samples were decellularized following as described[20,21]. In brief, cancer samples were decellularized in two rounds of lysis buffer [0.1% sodium dodecyl sulfate (SDS), 0.02% sodium azide, 5 mM EDTA, 0.4 mM phenylmethylsulfonyl fluoride (PMSF); all Sigma-Aldrich, St. Louis, MO, USA] for 6 hours followed by 15 min in rinse buffer (0.02% sodium azide, 5 mM EDTA, 0.4 mM PMSF). Then, PDSs were washed 72 hours in distilled water, renewed every 12 hours, and a last 24 hours wash in PBS (Medicago) to remove cellular debris. Decellularization was performed at 37 °C and agitation at 175 rpm (Incu-ShakerTM 10 L, Benchmark). PDSs were then placed in a storage solution containing 0.02% sodium azide and 5 mM EDTA in PBS at 4 °C to preserve the tissue until usage. To optimize the sample material use, PDSs were cut in 6 mm diameter and 150 µm thickness slices using a biopsy punch needle and a CM3050-S cryotome (Leica). PDS slices were sterilized in peracetic acid 0.1% (Sigma-Aldrich), 1 hour at room temperature, followed by 24 hours wash in PBS with 1% Antibiotic-Antimycotic (Thermo Fisher Scientific), at 37 °C and gentle agitation (175 rpm; Incu-ShakerTM 10 L, Benchmark).

## Patient-derived scaffolds recellularization and co-cultures

PDS slices were placed in a 48-well plate and $3 \times 10^5$ MCF-7 or MDA-MB-231 cells were added in 500 µl of cell line specific medium. After 24 hours, PDSs were transferred to a new plate with fresh media. This process was repeated once or twice a week depending on the cell growth rate, and incubation was continued for 21 days.

For co-cultures of cancer cells and T cells, PDSs were repopulated with MCF-7/Luciferase cancer cells for 21 days, as described above. Then, $2 \times 10^5$ non-activated or activated T cells were added and cultivated together 24 (mild activation) or 48 (strong activation) hours depending on the experiment. Afterwards, the medium containing the T cells was recovered, and after centrifugation, T cells were characterized by flow cytometry and the released molecules measured in the medium. MCF-7/Luciferase cells were detached from PDSs with Trypsin-EDTA (0.25%, Gibco, Fisher Scientific) and viability and changes in gene expression were analyzed. 2D monocultures and co-cultures in 24-well plates were performed in parallel to the PDS cultures and used as a reference for gene expression analyses and survival assays.

## Survival assays

Cancer cells were detached from PDSs or plate surface by 7 min incubation in Trypsin-EDTA (0.25%, Gibco, Fisher Scientific) at 37°C and agitation at 180 rpm. After Trypsin inactivation and removal, cells were resuspended in 350 µL supplemented RPMI-1640 medium, 250 µL were used for RNA extraction and gene expression analysis and 100 µL for measurement of viable cells by Luciferase luminescence assay. Serial dilutions of the cells were dispensed in 96-well plates (flat bottom clear, white polystyrene, Merck), and settled overnight (37°C, 5% CO2 humidified atmosphere). 150 µg mL$^{-1}$ D-luciferin substrate (Perkin Elmer) was added immediately prior the light

emission measurement in a 1420 Multilabel Counter Victor3 (Perkin Elmer). Standard curve using MCF-7/Luciferase from fresh 2D cultures were included in each plate and used to calculate the number of cells by light emission unit.

## Flow cytometry analyses

Flow cytometry analyses of T cells were performed on single cells after excluding dead cells with Live/Dead fixable aqua dead cell stain kit (Molecular Probes). Data acquisition was performed on a LSRII flow cytometer (BD Biosciences), equipped with FACS Diva software (BD Biosciences) and analyzed using FlowJo software (TreeStar Inc). The following antibodies were used: CD3-APC/H7 (clone SK7; ref. 560176), CD4-AF700 (clone OKT4; ref. 56-0048-80; Invitrogen), CD8-BUV395 (clone RPA-T8; ref. 563796), CD25-APC (clone M-A251; ref. 561399), PD-1-BUV737 (clone EH12.1; ref. 612792) and CD69-PE (clone FN50; ref. 560968) (all BD Biosciences).

## Immunofluorescence

PDS cultures were harvested and fixed in 4% paraformaldehyde solution (Sigma-Aldrich) overnight at 4°C. For sectioning, they were dehydrated and paraffin-embedded, and 5 µm thick sections were obtained using a Microm Cool-cut (Thermo Fisher Scientific). Sections were deparaffinized, rehydrated, and subjected to antigen retrieval buffer treatment (10 mM Tris, 1 mM EDTA, 0.05% Tween20, pH 9.0) in a 2100 retriever pressure cooker (Aptum Biologics Ltd.). For immunostaining, the Opal 6-Plex Manual Detection Kit was used following manufacturer's instructions (Akoya Biosciences). In brief, sections were blocked 10 min in blocking buffer and subsequently incubated with the primary antibody for 1 hour followed by 10 min of incubation with Opal Ms/Rb Polymer antibody and 10 min with Tyramide-Fluorochrome conjugate (1:150) diluted in 1xPlus Amplification buffer, all performed at room temperature. Afterwards, slides were microwave treated in antigen retrieval pH 9.0 buffer and the immunostaining protocol repeated for each primary antibody. Finally, nuclei were stained with DAPI solution for 15 min and slides mounted with ProLong™ Glass Antifade Mountant (Invitrogen) and covered with a #1.5 coverslip. Sections were scanned with Metafer Slide Scanning Platform (Metasystems, Heidelberg Germany) with Axio Imager.Z2 Microscope, 20/0.8/air objective (Zeiss, Oberkochen, Germany) and SpectraSplit filter set (Kromnigon, Gothenburg, Sweden). Primary antibodies and reactive fluorophores were used in the following order: anti-PD-L1 (1:250; clone [28-8], ref. ab205921, Abcam) with Opal 690, anti-PD1 (1:500; clone EPR4877(2), ref ab137132, Abcam) with Opal 620, anti-PanCK (1:500, clone KRT/1877R, ref ab234297, Abcam) with Opal 520, and anti-CD3 (1:500, polyclonal, ref. GA503, Agilent Dako) with Opal 570.

## Gene expression analysis

RNA extraction from cancer cells was performed in 350 µL of RLT buffer and using the RNeasy Micro Kit including DNase treatment, following the manufacturer's instructions (Qiagen). When the entire PDSs was used, samples were homogenized using a stainless-steel bead in TissueLyzer II (Qiagen), 5 min × 2 at 25 Hrz, centrifuged at 14,000 rpm for 3 min and supernatants transferred to new tubes prior RNA extraction. RNA concentration was measured using NanoDrop (Thermo Fisher Scientific) and RNA quality was randomly assessed using a 5400 Fragment analyzer HS RNA Kit (15NT) (DNF-472).

50-400 ng of RNA in 20 µl reaction mix were transcribed using GrandScript cDNA synthesis kit, including RNA Spike II (all TATAA Biocenter) for RNA stability control in a T100 Thermal Cycler (BioRad) and using a temperature profile of 25 °C for 5 min, 42 °C for 30 min, 85 °C for 5 min followed by cooling to 4 °C. Quantitative PCR (qPCR) was performed on a CFX384 Touch Real-Time PCR Detection System (BioRad) using 1x SYBR GrandMaster Mix (TATAA Biocenter), 400 nM primer mix (Supplementary Table 13) and 2 µl of diluted cDNA (1:5 or 1:6 dilution in RNAse-free water) in a final reaction volume of 6 µl. The temperature profile was 95 °C for 2 min followed by 35–50 cycles of amplification at 95 °C for 5 s, 60 °C for 20 s, and 70 °C for 20 s and a melting curve analysis at 65 °C

to 95 °C with 0.5 °C s$^{-1}$ increments. Cycle of quantification (Cq) values were determined by the second derivative maximum method with the CFX Manager Software version 3.1 (BioRad). Data preprocessing was performed using GenEx (MultiD). Cut-off for Cq values was established at 35 and missing data were replaced with a higher Cq value (Cq = 36). Gene expression was normalized using reference genes identified with the NormFinder algorithm and expressed as relative quantities (log2-scale) to control samples. All experiments were conducted in accordance with the MIQE guidelines[79].

### ELISA analyses
IFN-γ and Granzyme B released in the medium were measured using Human IFN-γ ELISA Kit and Human Granzyme B ELISA Kit respectively (Invitrogen), following the manufacturer's instructions.

### Mass spectrometry analysis
Available mass spectrometry data from 59 cell-free PDSs[37] were used to perform differential protein composition when stratifying PDSs based on their high and low *PD-L1* expression inducing capacity in MCF-7 cancer cells. To identify significantly differentially expressed proteins between both groups, Welch t-test was performed in SPSS statistics v.25 (IBM) and samples were ranked based on *p*-value. Log$_2$-fold change of the means (*PD-L1* high group vs. *PD-L1* low group) were calculated and visualized using a Volcano plot (Prism 10, GraphPad) and *p*-value = 0.05 as cut off. Pathways and processes enrichment were analyzed using ShinyGO 0.80 (http://bioinformatics.sdstate.edu/go/). Enrichment for transcription factor targets was analyzed using TRRUST database[80] and Metascape (https://metascape.org).

### Statistics and reproducibility
Principal component analyses were performed in GenEx (MultiId). Statistical analyses were performed in SPSS statistics v.25 (IBM) and Prism 10 (GraphPad). Experimental data are presented as median ± interquartile range (IQR) or mean ± SD, indicated in each case, and significant differences analyzed using unpaired t test, two-tailed. Mann-Whitney U and Kruskall-Wallis statistical tests were performed for assessment of clinico-pathological and molecular parameters. Correlations between gene expressions were analyzed using Pearson and Spearman's correlation coefficients. The Kaplan–Meier method was used to estimate disease-free survival (DFS) using log-rank comparisons in different gene expression strata divided by median, first quartile (25%) or third quartile (75%). *P*-values < 0.05 were considered significant. For gene expression studies and mass spectrometry, 110 and 59 frozen tumors from the Biobank were used, respectively. For PDS co-cultures with T cells, 17 tumors were used to generate the PDSs including at least 5 in each experiment. Replicates were defined as slices from the same PDS.

### Reporting summary
Further information on research design is available in the Nature Portfolio Reporting Summary linked to this article.

### Data availability
The data supporting the findings of this study are available within the paper, Supplementary Information and Supplementary Data 1 and 2 files. The mass spectrometry data that support the findings have been deposited to the ProteomeXchange Consortium (https://www.proteomexchange.org) via the PRIDE partner repository with the dataset identifier PXD042671[37].

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

## Acknowledgements
The authors thank Maria del Carmen Leiva Arrabal, Patrik Sundström and Pernilla Gregersson for technical assistance. Further, the authors would like to thank the Pathology and Surgical Department at Sahlgrenska University Hospital and Research Institutes of Sweden for their kind contributions to the project. This work was supported by grants from Swedish Research Council (2021-01008, 2019-01273), Swedish Cancer Society (22-2080, 22-2214), Sweden's Innovation Agency (2017-03737), the Swedish state under the agreement between the Swedish government and the county councils, the ALF-agreement (965065, 965580).

## Author contributions
Conceptualization: E.G., G.L. Formal analysis: E.G., S.R., A.G. Methodology: E.G., S.R., A.G., T.F.A., G.L., A.S., M.Q.J., L.S. Investigation: E.G., S.R. Validation: E.G., G.L. Resources: G.L., A.S., M.Q.J., K.H. Data curation: E.G., S.R., A.G. Writing original draft: E.G. Writing, review and editing: E.G., S.R., A.G., T.F.A., M.Q.J., L.S., K.H., G.L., A.S. Visualization: E.G., A.G. Supervision: G.L. Project administration: G.L., A.S. Funding acquisition: G.L., A.S. All authors have read and agreed to the published version of the manuscript.

## Funding

## Competing interests
The authors declare the following competing interests: Göran Landberg and Anders Ståhlberg are board members and shareholders of Iscaff Pharma AB. Anders Ståhlberg is a board member and shareholder in SiMSen Diagnostics and Tulebovaasta. The other authors declare no competing interests.
