## [Transparent Peer Review file · Communications Biology]

Breast cancer patient-derived scaffolds enhance the understanding of PD-L1 regulation and T cell cytotoxicity.

Corresponding Author: Ms Elena Garre

Version 0:

Reviewer comments:

Reviewer #1

(Remarks to the Author)

The authors demonstrate the use of patient-derived scaffolds from 110 breast cancer patients, grown with the breast cancer cell line MCF7, as an elaborate 3D model to represent the heterogeneity of the tumour microenvironment and variation in immune regulators between individual patients. Major findings include the variation of PD-L1 expression between patients correlated with high grade tumours and ductal type tumours, in addition to correlating to specific protein expression in individual samples. Variations in PD-L1 expression were also associated with proteins involved in a variety of T-cell functions.

Furthermore, the authors show the novel finding that the addition of strongly activated T-cells to these cultures results in the killing of mostly differentiated cancer cells, with remaining populations expressing markers for stemness, pluripotency, epithelial-to-mesenchymal transition and elevated PD-L1 expression. Use of the PD-1 blocker, Pembrolizumab, illustrated that patient samples with lower responses to T cell killing displayed higher expression of SOX2 and lower MKI67 expressions, highlighting resistance to T cell killing in pluripotent, non-proliferative cancer cells. Overall, these findings present a link between induction of PD-L1 expression, cancer stemness and resistance to T cells.

Overall, this report was well written and the topic was interesting. The authors provide strong evidence for their findings with relevant controls used throughout. The methodology is detailed and provides sufficient information for these experiments to be reproduced. The data advances our understanding of T cell and cancer cell interactions in 3D cultures, and the potential role of the PD-L1/PD1 axis in modulating these interactions. The results are also discussed fairly in terms of current literature, highlighting both supporting and conflicting literature and potential reasons for differences in findings.

The techniques used will be of interest to the field, where 3D models are useful for screening immune-based therapeutics in a heterogeneous cell environment, a powerful tool in comparison to 2D monolayer cultures and animal models that do not always fully recapitulate aspects of human diseases. In addition, patient-derived scaffolds, permit the dynamic study of cancer and T-cell interactions, as well as the use of individual patient-derived scaffolds to tailor treatments to individual patients. This manuscript will be of importance to scientists investigating the role of cancer cell evasion of the immune system, where 3D models are currently lacking or in optimisation stages.

Suggested revisions :

1. In the results section (page 4), initially both MCF7 and MDA-MB-231 cells were used. However, the rest of the paper only discusses results using MCF7 cells. Please can the authors add a statement to clarify the reasons for only proceeding to use MCF7 cells? Is this because of the negative or lack of upregulation of PD-L1 expression on MDA-MB-231/PDS cultures compared to MDA-MB-231 2D cultures as shown in supplementary Figure 1?
2. In Figure 2(f) a graph is not displayed for MYH11 although on page 7 a reference is made to this figure showing profiles for the 10 most differentially expressed proteins. Please include the graph.
3. On page 9 the decrease in the number of MCF-7 cells in PDS co-cultures related to activation grade is stated as 58% and 28% for mild/strong activated T cells but in supplementary summary table 7 it appears to be 53% and 27%. Please clarify if this is correct.
4. Figures 4(a-d) (page 11). Are there any supplementary data to show the statistical results shown?
5. Figure 4(e) (page 11) could be improved by showing images for individual markers stained to complement the merged

images shown. A control using no primary antibody should be included to show that the staining is shown not a result of non-specific background staining. Please also add scale bars.

6. On page 13 in the reference for Supplementary Figure 5, the data for CD24 is not displayed. Please show the data

7. Figure 5(e-f) could benefit from clarifying the criteria for selecting "responders" and "non-responders". For example, on page 16, PDS1 seems to be categorised as a "non-responder" but could be interpreted as a responder based on the graph in Figure 5(d).

8. Statistical analysis (page 25) - Could you clarify why the median was used as an average instead of mean averages throughout the paper?

Reviewer #2

(Remarks to the Author)

This is an interesting and informative paper. With minor questions (see below) the manuscript is carefully written and the data well presented. The findings are of importance given the now widespread used of immunotherapies and their variable inter-patient responses observed. In this context, the use of human tumour cell-free scaffolds (PDSs) will be of interest to the field. The authors should be commended for generating a large number of these PDSs.

Minor Comments

1: At the beginning of the results the authors say they studied MCF-7 and MDA-MB-231 cells but there is no further mention of the MDA-MB-231 cells in the main text. The only data presented is in Supplementary Fig. 1 where the author says "PD-L1 and PD-L2 expression induced triggered by the PDSs was detected in both cell lines, ..." However, SFig. 1 shows downregulation of PD-L1 and PD-L2 expression. I am confused - please can the authors clarify

2: Fig. 2a - Why were so few samples assessed for PD-L2 expression. The legend says n=110 but the number of data points is not 110. The legend should clarify this. Also, if PD-L2 expression was undetectable in 2D MCF-7 cells, how can these data be relative expression?

3: Fig. 3a - would be helpful if the headings CD4+ CD8- T cells and CD4- CD8+ T cells were larger.

4: A general point. At present immunotherapies are primarily used to treat ER- breast cancers. The majority of the PDSs here are from ER+ patients and MCF-7 are an ER+ cell line. As the authors say at the end of their introduction that the PDS system has the potential to evaluate immune response and potentially predict immunotherapy effects - the authors should discuss their choice of model in the manuscript and what their findings mean for ER+ breast cancer patients.

Version 1:

Reviewer comments:

Reviewer #1

(Remarks to the Author)

I am satisfied that revision points raised in my previous comments have been addressed by the authors and I have no further comments. The revised manuscript is ready to be submitted.

Reviewer #2

(Remarks to the Author)

I was reviewer 2 originally. In this 2nd submission manuscript the authors have satisfactorily addressed all of the comments I raised and revised their manuscript accordingly. No further comments.

POINT-BY-POINT RESPONSE TO THE REVIEWERS.

Reviewer #1 (Remarks to the Author):

The authors demonstrate the use of patient-derived scaffolds from 110 breast cancer patients, grown with the breast cancer cell line MCF7, as an elaborate 3D model to represent the heterogeneity of the tumour microenvironment and variation in immune regulators between individual patients. Major findings include the variation of PD-L1 expression between patients correlated with high grade tumours and ductal type tumours, in addition to correlating to specific protein expression in individual samples. Variations in PD-L1 expression were also associated with proteins involved in a variety of T-cell functions.

Furthermore, the authors show the novel finding that the addition of strongly activated T-cells to these cultures results in the killing of mostly differentiated cancer cells, with remaining populations expressing markers for stemness, pluripotency, epithelial-to-mesenchymal transition and elevated PD-L1 expression. Use of the PD-1 blocker, Pembrolizumab, illustrated that patient samples with lower responses to T cell killing displayed higher expression of SOX2 and lower MKI67 expressions, highlighting resistance to T cell killing in pluripotent, non-proliferative cancer cells. Overall, these findings present a link between induction of PD-L1 expression, cancer stemness and resistance to T cells.

Overall, this report was well written and the topic was interesting. The authors provide strong evidence for their findings with relevant controls used throughout. The methodology is detailed and provides sufficient information for these experiments to be reproduced. The data advances our understanding of T cell and cancer cell interactions in 3D cultures, and the potential role of the PD-L1/PD1 axis in modulating these interactions. The results are also discussed fairly in terms of current literature, highlighting both supporting and conflicting literature and potential reasons for differences in findings.

The techniques used will be of interest to the field, where 3D models are useful for screening immune-based therapeutics in a heterogeneous cell environment, a powerful tool in comparison to 2D monolayer cultures and animal models that do not always fully recapitulate aspects of human diseases. In addition, patient-derived scaffolds, permit the dynamic study of cancer and T-cell interactions, as well as the use of individual patient-derived scaffolds to tailor treatments to individual patients. This manuscript will be of importance to scientists investigating the role of cancer cell evasion of the immune system, where 3D models are currently lacking or in optimisation stages.

Suggested revisions:

1. In the results section (page 4), initially both MCF7 and MDA-MB-231 cells were used. However, the rest of the paper only discusses results using MCF7 cells. Please can the authors add a statement to clarify the reasons for only proceeding to use MCF7 cells? Is this because of the

negative or lack of upregulation of PD-L1 expression on MDA-MB-231/PDS cultures compared to MDA-MB-231 2D cultures as shown in supplementary Figure 1?

Response 1: As demonstrated in our previous work (Garre et al., 2022), different breast cancer cell lines exhibit both similar and distinct responses when adapted to PDS cultures. These variations are likely due to differences in genetic abnormalities and the differentiation status of the cancer cells, which influence their responses to varying cancer microenvironments. For instance, some cell lines, such as the undifferentiated triple-negative MDA-MB-231, have limited capacity to respond to the scaffold environment. In contrast, the more differentiated MCF-7 cell line showed greater responsiveness to the individual TMEs provided by the PDSs. Additionally, changes in PD-L1 gene expression in the PDS cohort cultured with MCF-7 cells were highly dependent on the PDS and linked to clinical variables and behaviours of the original cancer disease. Therefore, we concluded that MCF-7 is a more suitable reporter cell line for further experiments with T cells, as it can better reveal the information retained in the TME provided by the individual PDSs. This clarification has been included in the Results section.

Lines 108-112: “Remarkably, the expression levels of PD-L1 and PD-L2 varied substantially between PDS cultures, with the MCF-7 cell line exhibiting more pronounced differences compared to the triple-negative MDA-MB-231. This highlights a regulatory function for the human-based growth conditions, further influenced by the cancer cell line’s response to microenvironmental cues (Supplementary Table 3)”.

Lines 134-137: “Since the MDA-MB-231 cell line showed less responsiveness to the individual PDSs compared to MCF-7 cells, and the gene changes were not associated with clinical observations (Supplementary Table 4), we focused on the more informative MCF-7 cancer cell line for the upcoming PDS experiments”.

2. In Figure 2(f) a graph is not displayed for MYH11 although on page 7 a reference is made to this figure showing profiles for the 10 most differentially expressed proteins. Please include the graph.

Response 2: MYH11 is among the 10 most differentially expressed proteins (Supplementary Table 6) but is not significantly correlated with PD-L1 expression changes in MCF-7. Only the significant proteins have been included in Figure 2f.

The text in the Results section (line 165: “...9 out of the 10 most differentially expressed proteins correlated...”) and the figure legend (line 152: “Scatter plots illustrating significant correlations...”) have been modified for clarity.

A Supplementary Figure 3 with MYH11 has also been included.

Supplementary Figure 3. Scatter plot illustrating the correlation between MYH11 protein levels and PD-L1 mRNA levels induced in MCF-7 cancer cells.

3. On page 9 the decrease in the number of MCF-7 cells in PDS co-cultures related to activation grade is stated as 58% and 28% for mild/strong activated T cells but in supplementary summary table 7 it appears to be 53% and 27%. Please clarify if this is correct.

Response 3: Thank you for pointing out this error. The data in Figure 4 has been revised and the calculations have been repeated to verify the numbers. The corrections are presented in the Results section (lines 219-220) and in Supplementary Table 7. Additionally, numerical values have been included as supplementary material (Data Figure 4).

Lines 219-220: "53% and 27% median values of surviving cells with mild and strong activation respectively".

4. Figures 4(a-d) (page 11). Are there any supplementary data to show the statistical results shown?

Response 4: Statistical results and numerical data for the calculations are included in supplementary material (Data Figure 4).

5. Figure 4(e) (page 11) could be improved by showing images for individual markers stained to complement the merged images shown. A control using no primary antibody should be included to show that the staining is shown not a result of non-specific background staining. Please also add scale bars.

Images showing the individual markers are included in Figure 4e, with scale bars added.

When comparing PDS samples with MCF-7 cells cultured with and without activated T cells, we confirmed that there was no non-specific staining in the CD3 or PD-L1 channels when T cells were absent, indicating that the primary antibodies did not generate non-specific background (Fig. 4e). Additionally, the CD3 and pan-CK primary antibodies used in this study have been routinely used by the co-authors, and their specificity has been validated in previous studies (Rodin et al., PMID: 38953978; Andric et al., PMID: 36900249).

6. On page 13 in the reference for Supplementary Figure 5, the data for CD24 is not displayed. Please show the data

Response 6: CD24 gene expression has been added to the Supplementary Figure 6 (Note: the addition of supplementary figures has changed the number annotation).

Supplementary Figure 6. Relative expression of genes associated to cancer processes (differentiation – Diff., pluripotency, epithelial-to-mesenchymal transition – EMT, proliferation, cancer stem cell - CSC; apoptosis – Apt.) in MCF-7 PDS monocultures and co-cultures with strongly activated T cells where T cells were depleted from the cell sample immediately before RNA extraction using Dynabeads™ CD3 (Invitrogen). Average and standard deviation is plotted (replicates=4). Gene expression is expressed relative to 2D MCF-7 cultures and in log-2 scale.

7. Figure 5(e-f) could benefit from clarifying the criteria for selecting “responders” and “non-responders”. For example, on page 16, PDS1 seems to be categorised as a “non-responder” but could be interpreted as a responder based on the graph in Figure 5(d).

Response 7: Responders were defined as PDS co-cultures of cancer cells and T cells that showed at least a 10% reduction in viability when treated with Pembrolizumab compared to untreated PDS co-cultures. This clarification has been included in the Results section (lines 339-343), and the data has been added to the supplementary material (Data Figure 5).

Lines 339-343: “Two PDS cultures, PDS4 and PDS10, were clearly defined as “responders,” showing a 50-60% additive reduction in cancer cell viability. Three additional “responders,” PDS2, PDS3, and PDS8, exhibited slightly lower treatment effects with viability reductions of 28%, 26%, and 11%, respectively. The remaining five PDS cultures showed only minor effects, with less than a 10% reduction in viability, and were classified as “non-responders” (Fig. 5d)”.

8. Statistical analysis (page 25) - Could you clarify why the median was used as an average instead of mean averages throughout the paper?

Response 8: The median represents the central tendency of a data set and is used when the data is not expected to follow a normal distribution. Therefore, we used median values when comparing measurements from a group of PDSs, as we did not expect a normal distribution or similar values due to the PDSs originating from different patients and cancer microenvironments. This approach was applied to: a) gene expression measurements from a PDS cohort (Figure 1, Figure 5e-f), and b) viability and PD-L1 expression in PDS co-cultures (Figure 4). When measurements were performed on samples originating from the same PDS, the same T cell donor, or experimental replicates, we used mean values due to the less variable data.

Reviewer #2 (Remarks to the Author):

This is an interesting and informative paper. With minor questions (see below) the manuscript is carefully written and the data well presented. The findings are of importance given the now widespread use of immunotherapies and their variable inter-patient responses observed. In this context, the use of human tumour cell-free scaffolds (PDSs) will be of interest to the field. The authors should be commended for generating a large number of these PDSs.

Minor Comments

1: At the beginning of the results the authors say they studied MCF-7 and MDA-MB-231 cells but there is no further mention of the MDA-MB-231 cells in the main text. The only data presented is in Supplementary Fig. 1 where the author says “PD-L1 and PD-L2 expression induced triggered by the PDSs was detected in both cell lines, ...” However, SFig. 1 shows downregulation of PD-L1 and PD-L2 expression. I am confused - please can the authors clarify

Response 1: Thank you for pointing this out. The sentence and message have now been revised in the Results section (lines 105-107), to better depict the PD-L1 and PD-L2 expression in PDS cultures using both cell lines. The reason for the brief mention of MDA-MB-231 has been detailed above in the response to Comment 1 Reviewer 1.

Lines 105-107: “PD-L1 and PD-L2 expression were detected in both cell lines propagated in PDS cultures. Notably, 15 MCF-7 PDS cultures induced PD-L2 expression despite its absence in MCF-7 cells grown in 2D cultures”.

2: Fig. 2a - Why were so few samples assessed for PD-L2 expression. The legend say n=110 but the number of data points is not 110. The legend should clarify this. Also, if PD-L2 expression was undetectable in 2D MCF-7 cells, how can these data be relative expression?

Response 2: PD-L2 expression was assessed in 110 PDS cultures using MCF-7 cells and in 84 PDS cultures with MDA-MB-231 cells. However, only 15 PDS MCF-7 cultures showed detectable expression of PD-L2 gene, as noted on (lines 105-107). When using GenEx (MultiD) software for processing gene expression data, a cut-off of the Cq (cycle of quantification) minimal value to be considered as one cDNA copy is defined, with higher Cq value assigned to empty wells. This data preprocessing allows for the calculation of relative expression for samples transitioning from undetectable to detectable expression. A sentence explaining this data preprocessing has been included in Methods section (lines 597-598).

Lines 597-598: “Cut-off for Cq values was established at 35 and missing data were replaced with a higher Cq value (Cq = 36)”.

3: Fig. 3a - would be helpful if the headings CD4+ CD8- T cells and CD4- CD8+T cells were larger.

Response 3: Figure 3a has been revised and the headings have been made larger.

4: A general point. At present immunotherapies are primarily used to treat ER- breast cancers. The majority of the PDSs here are from ER+ patients and MCF-7 are an ER+ cell line. As the authors say at the end of their introduction that the PDS system has to potential to evaluate immune response and potentially predict immunotherapy effects - the authors should discuss their choice of model in the manuscript and what their findings mean for ER+ breast cancer patients.

Response 4: This is a relevant comment and an explanation of the choice of model has been added to the Results section (lines 134-137). Additionally, a paragraph discussing the relevance of the PDS model for assessing the immune response in ER-positive breast cancer has been included in the Discussion section (lines 438-450).

Lines 134-137: “Since the MDA-MB-231 cell line showed less responsiveness to the individual PDSs compared to MCF-7 cells, and the gene changes were not associated with clinical observations (Supplementary Table 4), we focused on the more informative MCF-7 cancer cell line for the upcoming PDS experiments”.

Lines 438-450: “Given that a high percentage of the PDSs in this study originated from estrogen receptor-positive breast cancers, our data support previous studies indicating the relevance of the

tumor microenvironment and immunological state in hormone-positive breast cancer. This subtype may potentially benefit from immunotherapies within an individualized, predictive treatment strategy 75, 76. Although, estrogen receptor-positive breast cancer has not traditionally been considered immunogenic, varying levels of tumor-infiltrating lymphocytes (TILs) have been observed and linked to prognostic values 77. Patients with aggressive metastatic hormone-positive breast cancer, have also been reported to respond to immune-checkpoint inhibitor treatment, although the effectiveness remains unclear 75, 78. Consistent with this, our results using PDS-cultures indicated that some estrogen receptor-positive breast cancer microenvironment, including an estrogen receptor breast cancer cell line, were sensitive to Pembrolizumab. This provides an additional tool for evaluating and predicting patient-specific immunotherapy responses”.

REVIEWERS' COMMENTS:

Reviewer #1 (Remarks to the Author):

I am satisfied that revision points raised in my previous comments have been addressed by the authors and I have no further comments. The revised manuscript is ready to be submitted.

Reviewer #2 (Remarks to the Author):

I was reviewer 2 originally. In this 2nd submission manuscript the authors have satisfactorily addressed all of the comments I raised and revised their manuscript accordingly. No further comments.

Corresponding authors' response:

There are not further comments from the reviewers to address.